# Trafficking and Association of *Plasmodium falciparum* MC-2TM with the Maurer’s Clefts

**DOI:** 10.3390/pathogens10040431

**Published:** 2021-04-05

**Authors:** Raghavendra Yadavalli, John W. Peterson, Judith A. Drazba, Tobili Y. Sam-Yellowe

**Affiliations:** 1Department of Biological, Geological, and Environmental Sciences, Cleveland State University, Cleveland, OH 44115, USA; r.yadavalli99@vikes.csuohio.edu; 2Imaging Core Facility, The Cleveland Clinic, Cleveland, OH 44195, USA; petersj@ccf.org (J.W.P.); drazbaj@ccf.org (J.A.D.)

**Keywords:** *Plasmodium falciparum*, Maurer’s clefts, PfMC-2TM, protein trafficking, clinical biomarker, *Plasmodium* membrane proteins

## Abstract

In this study, we investigated stage specific expression, trafficking, solubility and topology of endogenous PfMC-2TM in *P. falciparum* (3D7) infected erythrocytes. Following Brefeldin A (BFA) treatment of parasites, PfMC-2TM traffic was evaluated using immunofluorescence with antibodies reactive with PfMC-2TM. PfMC-2TM is sensitive to BFA treatment and permeabilization of infected erythrocytes with streptolysin O (SLO) and saponin, showed that the N and C-termini of PfMC-2TM are exposed to the erythrocyte cytoplasm with the central portion of the protein protected in the MC membranes. PfMC-2TM was expressed as early as 4 h post invasion (hpi), was tightly colocalized with REX-1 and trafficked to the erythrocyte membrane without a change in solubility. PfMC-2TM associated with the MC and infected erythrocyte membrane and was resistant to extraction with alkaline sodium carbonate, suggestive of protein-lipid interactions with membranes of the MC and erythrocyte. PfMC-2TM is an additional marker of the nascent MCs.

## 1. Introduction

*Plasmodium falciparum* merozoites induce profound morphological changes in the host erythrocyte cytoplasm following invasion. The erythrocyte cytoplasm is remodeled by the parasite and results in formation of membranous structures called Maurer’s clefts [1]. Maurer’s clefts facilitate protein transport to the erythrocyte membrane, cytoskeleton and surface [2,3,4]. Parasite derived proteins are transported across the parasite membrane (PM) and parasitophorous vacuole membrane (PVM), to the erythrocyte cytoplasm and to the surface of the infected erythrocyte through the Maurer’s clefts [5,6]. On the erythrocyte surface, knobs are formed incorporating variable surface antigens (VSAs) such as the *var* gene product, *P. falciparum* erythrocyte membrane protein 1 (PfEMP1) [7] and the knob-associated histidine rich protein (KAHRP). The knobs mediate cytoadherance of the infected erythrocytes to endothelial receptors such as intracellular adhesion molecule 1 (ICAM-1), vascular cell adhesion molecule 1 (VCAM-1) and CD36 [8,9,10,11]. The *var* genes along with the *STEVOR*, *SURFIN* and *RIFIN* genes are members of large multicopy-gene families encoding variable proteins participating in infected red blood cell cytoadherance, sequesteration and rosetting [12,13]. A fifth multigene family encodes the *Plasmodium falciparum* Maurer’s clefts two transmembrane (PfMC-2TM) proteins, consisting of 13 members distributed on 8 chromosomes in *P. falciparum*. Protein sequence alignment shows 79% sequence identity among family members. The orthologues for the *PfMC-2TM* genes are present in *P. reichenowi* (CDC), *P. billcollinsi* (G01), *P. blacklocki* (G01) and *P. praefalciparum* (G01) (PlasmodB).

Some of the VSAs participating in knob formation associate with membranous structures in the Maurer’s clefts (MC). The ring exported protein1 (REX1) [13,14,15], skeletal binding protein 1 (SBP1) [16], trophozoite exported protein 1 (TEX1) [17], Pf332 [18], membrane associated histidine rich protein 1 and 2; MAHRP1 [19] and MAHRP2 [1] play varying roles in biogenesis, stabilizing and tethering of the clefts and also in facilitating PfEMP1 protein transit through the clefts to the erythrocyte surface [20,21,22,23]. PfMC-2TM possesses a canonical signal sequence that overlaps with the first transmembrane domain and possesses PEXEL and vacuolar translocation motifs for transport through the PVM [24]. Studies show that the PEXEL motif is essential for the proteins to be exported into erythrocyte cytoplasm. However, not all proteins trafficked to the erythrocyte cytoplasm are PEXEL dependent. Immunoelectron microscopy and colocalization studies with antibodies against REX-1 show that PfMC-2TM is localized to the PV membrane and to the Maurer’s clefts [24,25]. Following export into the erythrocyte cytoplasm, PfMC-2TM is shown to have varying topological organization within the MC membrane. Amino and carboxyl termini of PfMC-2TM have been reported oriented towards the erythrocyte cytoplasm with the TM domains associated with the MC membrane [26]. The orientation of the two termini was also shown to be towards the MC lumen [27] and the variable spacer between TM domains was shown to be exposed on the erythrocyte surface [12,28]. In other studies, PfMC-2TM was not detected on the surface of the infected erythrocyte [24,29].

Due to the conflicting data regarding the stage specific expression, traffic, topology and membrane orientation of PfMC-2TM in the MC and erythrocyte membrane, we performed a detailed cell biological and biochemical characterization of endogenous PfMC-2TM export in *P. falciparum*. Investigations of endogenous PfMC-2TM traffic from the parasite cytoplasm to the MC have not been performed. The mechanism of PfMC-2TM export into the erythrocyte cytoplasm is unknown. Similarly, protease accessibility studies to identify protein topology have also not been performed. We investigated PfMC-2TM expression following merozoite invasion and protein traffic through the parasite endomembrane system using Brefeldin A. Antisera prepared against PfMC-2TM peptides located at the N and C termini of the protein and a monoclonal antibody specific for an epitope in the central portion of PfMC-2TM, were used to identify the orientation of protein. The topological association of PfMC-2TM with the MC and erythrocyte membrane was investigated using streptolysin O (SLO) and detergent permeabilization of the infected erythrocyte and parasite membranes. Alkaline sodium carbonate fractionation and protease accessibility studies were performed to identify the topological organization of the protein in the erythrocyte cytoplasm, MC and infected erythrocyte membrane.

## 2. Results

### 2.1. In Silico Analysis Predicts PfMC-2TM to Possess 3 Transmembrane Domains

Protein sequence alignments of the PfMC-2TM orthologues are shown in yellow and blue highlighted areas of Figure 1A. The sequence shows an overall 79% similarity among the sequences from *P. falciparum* strains 3D7 and IT and sequences from *P. reichenowi* (PlasmoDB & OrthoMCL). Twelve of the 13 PfMC-2TM multifamily proteins were predicted to possess three transmembrane domains with the signal peptide overlapping the first transmembrane domain at the N-terminus of the proteins from *P. falciparum*. In silico analysis from PlasmoDB (http://plasmodb.org, accessed on 15 February 2021), shows that the predicted protein PfMC-2TM expressed by PF3D7_0114100 has a length of 229 amino acids (aa) and contains 3 transmembrane domains (Figure 1B). The predicted position of the signal peptide (SP) spans amino acid position 1 to 12, transmembrane domain 1 (TM1), spans amino acid position 4 to 26, transmembrane domain 2 (TM2) from position 157 to 179 and transmembrane domain 3 (TM3) from position 184 to 203 (Figure 1B). Domains were evaluated using PlasmoDB and HMMTOP analysis. These TM domains are represented as hydrophobic residues, with the remaining portions of the protein as polar. The predictions of hydropathy plot and secondary structure of the sequence suggests the protein possesses three helical structures at individual TM domain regions (PlasmoDB). Two of the three transmembrane domains in PfMC-2TM; from position 157 to 179 and 184 to 203 have been identified to be associated with the PVM and MC membrane previously [24]. The originally annotated N-terminus of PfMC-2TM was shown to have a signal peptide (SP) overlapping with TM1 but it was not cleaved upon protein translocation [24,30]. The topology of TM1 domain at the N-terminus spanning amino acids 4 to 26 and its association with the MC is unknown. PfMC2TM is an integral membrane protein that possesses a PEXEL motif. Secondary structure analysis of the protein shows that the protein possesses four helices with two helices predicted at the N terminus and two at the C terminus of the protein (PlasmoDB).

### 2.2. PfMC-2TM Is Expressed by 4 h Post Merozoite Invasion

In order to identify the timing of PfMC-2TM protein expression following merozoite invasion into the host erythrocyte, stage specific parasites were collected and protein expression identified using IFA. PfMC-2TM was detected at 4 hpi in ring stage parasites tightly colocalized with REX1 (Figure 2A). PfMC-2TM and REX1 were localized to punctate structures in the erythrocyte cytoplasm. Protein expression progressed through the ring, trophozoite and schizont stages with localization to cytoplasmic MC distributed within the erythrocyte cytoplasm. At 8 and 12 hpi (early ring/mid ring stages) (Table 1), PfMC-2TM was detected within the parasite proximal to the nucleus and also in the erythrocyte in cytoplasmic MC. At 16 hpi and 30 hpi, (late ring to trophozoite stage of the parasite), PfMC-2TM was mostly exported to the host cytoplasm of the infected RBC, with the MC localized to the periphery of the erythrocyte membrane. In rings and trophozoites, PfMC-2TM was mostly localized in the erythrocyte cytoplasm suggestive of the association with PVM and MCs. In the late schizont stage, antibody reactivity was observed to be diffuse in the erythrocyte cytoplasm and by the segmented schizont stage at 51 hpi, the punctate MC were not distinguished as the MC appeared to flatten within the erythrocyte cytoplasm. PfMC-2TM protein expression progressed throughout the intra-erythrocytic cycle.

Protein expression in 4hpi rings was detected colocalized with REX-1 specific antibody. In order to validate endogenous protein expression for the blood stage, expression levels of SERA1 and RhopH3 proteins, used as controls to verify parasite stages and expression of known proteins [31], were also analyzed at each time point to ensure the integrity of stage specific protein expression (Figure 2B). SERA1 and RhopH3 proteins used as positive controls for stage specific expression showed protein expression at each parasite stage as reported previously [31,32].

### 2.3. PfMC-2TM Is Exported Using the Classical Secretory Pathway

In order to investigate the route of PfMC-2TM transport from the endoplasmic reticulum and Golgi in the parasite cytoplasm to the erythrocyte cytoplasm following protein expression, parasites were treated with the reversible acting drug, Brefeldin A (BFA), a fungal metabolite. BFA disrupts the function of the Golgi apparatus by blocking the GTP-dependent interaction of ARF in a reversible manner, resulting in redistribution of proteins into the ER [33]. RhopH3 and SERA1 sensitivity to BFA in synchronized parasites was tested to validate known proteins trafficked in a BFA sensitive manner as positive controls for BFA treatments (Figure 3A). Antibodies specific to the N and C termini of PfMC-2TM (antiserum 703 and 704, respectively) showed similar reactivity to the monoclonal antibody in identifying BFA sensitivity of PfMC-2TM (Figure 3B). Untreated parasites and vehicle (ethanol) treated negative controls were also reacted with antibodies and showed normal trafficking of proteins. In order to determine the effects of BFA treatment on protein transport in ring and trophozoite stages and to identify compartments affected by arrested protein traffic, BFA treatments were carried out for 16 h (rings) and 12 h (trophozoites). When 3 h old ring stage parasites were treated with BFA for 16 h (Ring + BFA), PfMC-2TM accumulated within the parasite in a structure surrounding the parasite nucleus suggestive of the ER (Figure 3C). Reactivity of anti-MAHRP1was also evaluated in BFA treated rings (Figure 3C). Anti-PfMC-2TM monoclonal antibody reactivity with 3D7 parasites was colocalized with anti-REX1 (Figure 3C). Upon release of the drug pressure, PfMC-2TM distribution in the infected erythrocyte cytoplasm was observed localized to punctate MCs (Figure 3C). Reactivity of BFA treated parasites with anti-ERD2 confirmed the accumulation of proteins in the ER during drug treatment (Figure 3D). Antibodies against the Maurer’s cleft protein Pf130 [24] showed that in contrast to PfMC-2TM, Pf130 is insensitive to BFA treatment (Figure 3D). When 21 h old trophozoites were treated with BFA for 12 h (Trophs + BFA), PfMC-2TM accumulated in the parasite cytoplasm. In contrast, when the drug pressure was released after 12 h, PfMC-2TM became distributed in the erythrocyte cytoplasm in punctate MCs proximal to the erythrocyte membrane (Figure 4B). After treatment of 37 h old schizonts with BFA for 9 h, PfMC-2TM was observed to be accumulated in the erythrocyte cytoplasm. Six hours post release of the drug pressure, PfMC-2TM was distributed in the erythrocyte cytoplasm and in MC tethered to the erythrocyte membrane (Figure 4C). Similar to the BFA treatment of ring stages, an anti-ERD2 antibody [34] was used to verify the compartments containing PFMC-2TM in BFA treated parasites. Anti-ERD2 reactivity was detected surrounding the parasite nucleus and in the ER. Antibody reactivity to MAHRP1 also detected protein accumulation in the parasites treated with BFA (Figure 3D and Figure 4D). Antibodies against the N- and C-termini of PfMC-2TM and a monoclonal antibody against PfMC-2TM had similar reactivity to BFA treated ring, trophozoite and schizont stages demonstrating the sensitivity of PfMC-2TM to BFA treatment, thus confirming trafficking of PfMC-2TM through the classical secretory pathway (Figure 3B and Figure 4B). Untreated and vehicle treated parasite controls showed normal growth and viability and normal distribution of PfMC-2TM in cytosolic membranous structures and MC by IFA (Figure 5). Negative controls using normal mouse and rabbit serum showed no protein reactivity (Figure 5). These results show that export of PfMC-2TM to the erythrocyte cytoplasm is blocked by BFA, confirming protein export through the classical secretory pathway.

### 2.4. PfMC-2TM Is Associated with the MC Membrane with the N- and C-termini Exposed to the Erythrocyte Cytoplasm

We next investigated the topology of PfMC-2TM in the erythrocyte cytoplasm following the demonstration of BFA sensitivity by PfMC2TM. Infected erythrocytes were formalin-fixed and permeabilized with streptolysin O (SLO) alone, SLO in combination with saponin with (+) or without (−) treatment with 1 mg/mL trypsin. A schematic illustration showing targets of permeabilization by SLO, Saponin and trypsin treatments is shown in Figure 6. Smears of the permeabilized cells were prepared on glass slides and IFA studies were performed to confirm the distribution, localization and protease accessibility of PfMC-2TM. The anti-N-terminus of PfMC-2TM (703), anti-C-terminus of PfMC-2TM (704) and monoclonal antibody SP1C1 [24] were used for the analysis. SLO selectively forms pores on the RBC plasma membrane, leaving PVM and MC membranes intact. Saponin disrupts the erythrocyte plasma membrane, MC membrane and PVM, but leaves parasite membrane intact (Figure 6) [35,36,37]. The parasite membrane remains intact so protein in the parasite is detected in the pellet if the protein is still associated with the parasite membrane.

### 2.5. PfMC-2TM in the MC Is Accessible to Protease in the Erythrocyte Cytoplasm

When infected erythrocytes were permeabilized with SLO alone, anti-PfMC-2TM reactivity was obtained with cytoplasmic structures in the erythrocyte cytoplasm. Following treatment of SLO permeabilized erythrocytes with saponin, protein reactivity was still detected but weaker. With trypsin treatment or a combination of saponin and trypsin treatment, anti-PfMC-2TM peptide antibodies specific for the N and C termini of the protein no longer detected the protein (Figure 7A,B), demonstrating that the N and C-termini of PfMC-2TM were exposed to the erythrocyte cytoplasm and accessible to trypsin digestion. A monoclonal antibody against PfMC-2TM, specific for an epitope in the central portion of PfMC-2TM was also used to analyze the topology of PfMC-2TM following permeabilization and protease treatment. The antibody detected PfMC-2TM in the erythrocyte cytoplasm following treatments with SLO alone, saponin or trypsin alone or combined saponin and trypsin treatment (Figure 7A,B), suggesting that the central section of PfMC-2TM was protected in association with MC membranes. Antibodies against known positive control proteins of RhopH3 and SERA1 were used to confirm the integrity of erythrocyte, PV and parasite compartments following permeabilization. Antibodies against the high molecular weight RhopH3 rhoptry protein and Serine rich antigen 1(SERA1) were used as markers for the parasite and PV, respectively. Antibody reactivity with both proteins demonstrated that the integrity of the parasite and PV compartments following permeabilization and detergent treatments was intact (Figure 7C,D). We further detected REX1 in the erythrocyte cytoplasm following treatment with SLO, Sap, and trypsin alone. However, treatment with SLO + Sap + Trypsin led to decreased reactivity of REX1 in a localized section within the cytoplasm (Figure 7E). Normal mouse and rabbit serum used as negative controls showed no reactivity with parasite proteins (Figure 7F).

### 2.6. PfMC-2TM Remains an Integral Membrane Protein upon Association with the Erythrocyte Membrane

We investigated the solubility properties of PfMC2TM using Triton X-100, alkaline sodium carbonate, urea and EDTA. Triton X-100 solubilizes peripheral and transmembrane proteins, sodium carbonate at pH 11 extracts peripheral membrane proteins and leaves transmembrane proteins intact in the pellet, 8M urea is a chaotropic reagent that solubilizes and disrupts protein complexes and EDTA chelates divalent cations destabilizing membrane proteins. The different reagents were used to treat infected red blood cell membrane ghosts obtained after hypotonic lysis of red blood cells as well as parasite pellets in the presence of absence of proteases. PfMC2TM would be accessible to protease treatment in domains exposed to the erythrocyte cytoplasm. Retention of PfMC2TM in the infected membrane ghosts following alkaline carbonate extraction and treatment with urea would indicate that the protein retained its integral membrane properties and was insoluble, respectively. We hypothesized that PfMC- 2TM associates with the erythrocyte membrane in addition to the MC membranes using the 3TM domains and becomes inaccessible in the RBC membrane due to potential protein-lipid interactions. In order to verify PfMC-2TM topology in association with the erythrocyte membrane, membrane ghost fractions were prepared by hypotonic lysis using 10 mM Tris, pH 8.8 or diluted RPMI (1:5) and fractionated with chaotropic agents, detergents, alkaline sodium carbonate and mechanical disruption. When fractionated samples were analyzed by western blotting using PfMC-2TM specific antibodies, PfMC-2TM was partially extracted by triton × 100 (Appendix A, lane 1). PfMC-2TM was resistant to extraction by 8M urea (Appendix A, lane 2), suggesting that the protein is insoluble. Membrane ghosts prepared from infected erythrocytes and treated with 1 mg/mL trypsin showed that PfMC-2TM was accessible to trypsin (Appendix A, lane 3). The protein was accessible to protease digestion with or without triton treatment (Appendix A, lanes 3 and 4), suggesting that PfMC-2TM is exposed to the cytoplasmic side of the infected erythrocyte. PfMC-2TM was detected in the erythrocyte membrane following EDTA treatment alone or in combination with freeze–thaw cycles (Appendix A, lanes 5 and 6). PfMC-2TM was not detected in the supernatants (Appendix A, lanes 7–12). Association of PfMC-2TM with infected erythrocyte membrane was also evaluated (Figure 8). Antibodies to the erythrocyte cytoskeletal protein α-spectrin and antibodies against the Maurer’s cleft protein Pf130 were used for analysis. The membranes were also probed with anti-PfMC-2TM (SP1C1), anti-α-spectrin and anti-Pf130 antibodies. PfMC-2TM reactivity was reduced upon trypsin digestion. As a result, faint bands were observed in pellet and supernatant (Figure 8, lanes 1 and 2, anti-PfMC-2TM panel). PfMC-2TM was resistant to trypsin digestion in 3D7 parasite as the 27 kDa band was detected in pellets and supernatant (Figure 8, lane 3 and 4, anti-PfMC-2TM panel). The 27 kDa protein band was resistant to the Na_2_CO_3_ extraction in infected red blood cell membrane ghost. The protein was found in pellet and no band was observed in the supernatant (Figure 8 lanes 5 and 6 anti-PfMC-2TM panel).

The 27 kDa protein band was also resistant to the Na_2_CO_3_ extraction in infected red blood cell membrane ghost subjected to freeze–thaw and vortexing for mechanical disruption. The protein was found in the pellet and no band was observed in the supernatant (Figure 8 lanes 7 and 8, anti-PfMC-2TM panel). PfMC-2TM was resistant to Na_2_CO_3_ extraction in 3D7 parasites and the control pellets as the band was observed in the pellet but not in the supernatants, (Figure 8 lanes 9–12). A very faint band at 27 kDa region was observed in the supernatant (Figure 8 lane 12). Anti- α spectrin antibody used as the control showed that the 220 kDa protein remained in the pellets of membrane ghost, showing complete resistance to sodium carbonate extraction (anti-α-spectrin panel Figure 8).

We further wanted to identify if the protein solubility was changed during protein traffic. Membrane ghosts prepared from infected erythrocytes were treated with 100 mM sodium carbonate, pH 11. PfMC-2TM was resistant to carbonate extraction whether treated with carbonate alone or combined with freeze and thaw cycles (Appendix A, lanes 5 and 6). The protein remained in the infected erythrocyte pellet suggesting that PfMC-2TM retains its integral membrane property during traffic and association with the erythrocyte membrane (Appendix A), similar to PfMC-2TM association with the parasite pellet following hypotonic lysis of infected erythrocytes to obtain parasite pellets (Appendix A, lanes 9 and 10). No antibody reactivity was detected in control uninfected erythrocyte membrane ghosts (Appendix A, lanes 1–4).

## 3. Discussion

Maurer’s clefts are parasite derived membranous structures produced in the erythrocyte cytoplasm immediately following *P. falciparum* merozoite invasion into the erythrocyte host cell [38]. MC distribution varies among *P. falciparum* isolates leading to differences in the amounts of proteins detected en route to the clefts in the PVM, MC and erythrocyte membrane [39]. *Plasmodium falciparum* possesses multi-copy gene families such as *var*, *RIFIN*, *STEVOR*, *SURFIN* and *PfMC-2TM* which encodes for variant surface antigens that participate in knob formation, rosetting and cytoadherance and are trafficked through the MC alone or chaperoned by other parasite proteins that transiently associate with the MC [5,40]. Cytoadherence, sequestration and antigenic variation are examples of strategies employed by *P. falciparum* for evasion of spleen clearance and immune responses [41,42,43]. The function of PfMC-2TM multi-copy gene family members in pathogenesis and immune evasion are unknown. In the current study, we performed detailed biochemical and cell biological approaches to characterize endogenous PfMC-2TM association with the MC and erythrocyte and parasite membranes. Following invasion of the host erythrocyte by merozoites, the differentiated ring stage parasite becomes surrounded by the parasite membrane, PVM and erythrocyte membrane. The three membranes are traversed by exported and imported metabolites, nutrients and other cargo necessary for intracellular parasite survival. In this study, we used specific antibodies targeting different sections of the PfMC-2TM protein, to investigate the export, solubility and topology of PfMC-2TM in the parasite and erythrocyte cytoplasm and membranes. The subcellular localization and topological distribution of PfMC-2TM in the erythrocyte cytoplasm and erythrocyte membrane is controversial and in need of clarification.

The first goal of this study was to identify the stage specific expression and translocation of PfMC-2TM into the erythrocyte cytoplasm. We used rabbit antisera and monoclonal antibodies specific to PfMC-2TM in confocal microscopy to identify the stage specific expression of PfMC-2TM in the blood stage. Timing of peak transcript expression does not always correspond to protein expression. Available transcriptome data showed peak transcript expression 24 hpi (PlasmodB). Furthermore, [12] reported initial protein expression at 10.5 hpi in *P. falciparum* 3D7 strain, with two peaks of transcription observed in clinical isolates. We show in the current study that PfMC-2TM expression was detected as early as 4 hpi in agreement with the biogenesis of MC soon after ring stage formation. Expression at 4 hpi was detected with peptide specific antibodies and a monoclonal antibody specific for PfMC-2TM. Tight colocalization with REX1 specific antibodies and detection of MAHRP1 and SBP1 provides further corroboration for the early expression of PfMC-2TM in the blood stage. Protein expression may be present before 4 hpi [1]. However, in order to unambiguously detect the protein expressed early in the ring stage, parasite collection commenced at 4 hpi. Intense antibody reactivity was observed in close proximity to the parasite nucleus at 4 hpi. By 16 hpi the protein was found to be exported to the erythrocyte cytoplasm and the presence of punctate staining structures were observed in the erythrocyte cytoplasm. Using antibodies against REX1, tight colocalization of REX1 and PfMC2TM were observed at 4 hpi and through the progression of the blood stage to segmented schizonts at 51 hpi. Our data shows that PfMC-2TM along with REX1 and SBP1 [44], is an additional marker for the nascent MCs.

In order to understand if PfMC-2TM is synthesized and trafficked through the classical secretory pathway for association with the MC, tightly synchronized parasites were treated with BFA. At 3 hpi, ring stage parasites were treated with BFA for 16 h, resulting in accumulation of PfMC-2TM in the parasite. Upon removal of the drug, the protein was exported into the erythrocyte cytoplasm and associated with the MC. At 30 hpi, trophozoites treated with BFA showed protein accumulation in the parasite along with punctate staining of the MC in the cytoplasm. Protein accumulation was also observed around individual merozoites in the schizont stage of the parasite. These results confirm that the protein is transported to the host cell cytoplasm in the ring stage of the parasite. The accumulation of protein in close proximity to the nucleus upon treating with BFA indicates that the protein is accumulated in the ER which indicates that the protein is exported in the classical secretory pathway to the host cell cytoplasm. Several studies have employed long incubation times of ring and trophozoite stage parasites in BFA, for up to 20 h to demonstrate sensitivity of protein transport to BFA treatment [17,18,33,42,45]. In each case, protein transport was recovered after drug removal, with protein localization to target organelles and structures. Ref. [45] reported reversal of parasite arrest after 24 h incubation with BFA, with parasite growth and progression into the next cycle. The use of IFA, solubility studies or metabolic labeling with ^35^[S]-methionine is used to analyze effects of BFA treatment on protein transport [33,42,45]. In the current study, IFA was used to show accumulation of protein in the ER following BFA treatment. Anti-ERD2 reactivity colocalized with SP1C1, which verified that proteins were accumulated in the ER as a result of BFA treatment. In eukaryotes, protein secretion is classified into classical and non-classical pathways [17,33]. In the classical pathway, the proteins are synthesized by ribosomes bound to the ER while protein is translocated or inserted into the ER post translationally, followed by vesicular transport from the ER via Golgi to the cell surface or the extracellular spaces. In contrast, molecular mechanisms involved in the nonclassical pathway of protein traffic are independent of ER/Golgi as reviewed in [17]. PfMC-2TM is constitutively expressed and exported to the erythrocyte cytoplasm throughout the blood stage. PfMC-2TM possesses a signal peptide, three transmembrane domains and has a PEXEL/HT motif which designates the protein for transport across the PVM into the erythrocyte cytoplasm [24,30]. However, it is unknown if plasmepsin V cleaves the PEXEL motif of PfMC-2TM for its traffic to the MCs. Antibodies prepared to a decapeptide consisting of the first ten amino acids at the N-terminus of PfMC-2TM recognize mature PfMC-2TM [24]. Other transmembrane proteins and members of multigene families such as RIFIN, STEVOR, PfEMP1, KAHRP, REX-2, SBP-1 and RESA have been investigated using mutational analysis and complementation studies, to identify sequences that facilitate protein export from the parasite through the PVM to the erythrocyte cytoplasm [46,47,48,49]. However, PfMC-2TM has not been investigated to identify sequences required for export and the role of the three annotated transmembrane domains in PfMC-2TM including the role of amino acid sequences in the N and C termini of the protein for export are unknown. Additional experiments will be performed to gain insights into the mechanisms of processing and transport of PfMC-2TM through the PVM.

The second goal of this study was to verify the topological organization of PfMC-2TM on the MC and infected erythrocyte membrane. Previous studies showed that PfMC-2TM is associated with the MC and erythrocyte membrane with N and C- termini exposed to the erythrocyte cytoplasm [26,28]. Other studies showed the N and C- termini of PfMC-2TM in the lumen of MCs [27]. To verify if either the C-terminus or N-terminus or both of PfMC-2TM protein is facing the erythrocyte cytoplasm following its association with the MC membrane, we employed a biochemical approach based on accessibility of PfMC-2TM to trypsin digestion after selectively permeabilizing the infected erythrocyte membrane using streptolysin O (SLO) and saponin, digitonin and triton [24,35]. SLO selectively forms pores on the erythrocyte plasma membrane, leaving PVM and MC membranes intact [35,36,37]. Saponin disrupts the RBC plasma membrane, MC membrane and PVM, but leaves the parasite membrane intact. Digitonin and Triton on the other hand disrupt the parasite membrane allowing parasite cytoplasmic proteins to become accessible to trypsin and proteinase K digestion. Infected erythrocytes parasitized with late trophozoites and early schizonts were treated with SLO and then treated with (+) or without (−) 5 mg/mL of proteinase K in a protease protection assay. In contrast to previously reported data [3}, we show that infected membrane-associated PfMC-2TM was accessible to protease. Our results also show that the membrane-associated PfMC-2TM is resistant to urea and sodium carbonate extraction. Moreover, PfMC-2TM associated with the membrane was accessible to trypsin digestion, contrary to our hypothesis. These results also differ from reports of trypsin insensitivity obtained for membrane associated PfMC-2TM in other studies [28]. This data suggests the varying distribution and topology of PfMC-2TM in detergent resistant sub-domains of the membrane that may be accessible to protease treatment. The results do not reflect accessibility of PfMC-2TM in MC membranes associated with infected RBC membrane ghosts following hypotonic lysis, as PfMC-2TM would still be detected in the infected RBC membrane ghosts due to protection within the membrane following protease treatment. This data suggests the distribution of PfMC-2TM in detergent resistant sub-domains of the membrane. It is unknown if the interactions of PfMC-2TM with the membrane are protein-protein or protein-lipid. PfMC-2TM was both triton and urea insoluble and did not change solubility in transit to the erythrocyte membrane. Furthermore, PfMC-2TM was identified by antibodies in punctate MC distributed in the erythrocyte cytoplasm and in the periphery of the infected cell proximal to the membrane in late trophozoites and schizont stages. Variants of PfMC-2TM family members may be differentially expressed and show different levels of reactivity with the antibodies used. This may account for the differences observed in both studies. How PfMC-2TM is translocated across the PVM to become associated with the MCs and erythrocyte membrane and surface is currently not known. Whether PfMC-2TM associates with RhopH3 in the PVM, MC and erythrocyte membrane is also unknown. RhopH1/Clag a member of the high molecular weight rhoptry complex along with RhopH3 and RhopH2, has been shown to participate in the formation of new permeability pathways (NPPs) in the infected erythrocyte [50,51]. One of the transporters formed in the infected erythrocyte is the *Plasmodium* surface anion channel (PSAC) shown to be important in nutrient transport. It will be important to determine if individual members of the PfMC-2TM family associate with PSAC or other transporters in the parasite membrane, PVM or erythrocyte membrane. The annotation of three TM for PfMC-2TM if accurate, may necessitate a change of the 2TM designation for the protein with a topology model for the three TMs as shown in Figure 9. A role for PfMC-2TM in the export of virulence markers to the erythrocyte surface or for macromolecular transport will define PfMC-2TM as an important therapeutic or vaccine candidate for malaria.

## 4. Materials and Methods

### 4.1. Plasmodium Falciparum Cultures

*Plasmodium falciparum* 3D7 strain was cultured according to previously described methods [52]. Parasites were grown in type A+ human erythrocytes (Interstate Blood Bank, Memphis, TN) at 5% hematocrit in RPMI-1640-HEPES complete media supplemented with 10% A+ human serum (Interstate Blood Bank) and 20 mM glucose. Schizont- infected erythrocytes were synchronized using a 65% Percoll gradient [53] and ring-infected erythrocytes were synchronized using 5% sorbitol lysis of mature parasites [54].

### 4.2. Stage Specific Protein Expression

Plasmodium. falciparum (strains 3D7)—infected erythrocytes were synchronized using 65% percoll to obtain 80% mature segmented schizonts as described previously [31]. Concentrated segmented schizonts were mixed with uninfected erythrocytes to obtain a parasitemia of 3% schizonts for merozoite invasion. Five to 6% parasitemia was obtained at reinvasion as shown in Table 1. Parasite-infected erythrocytes were collected at 4, 8, 12, 16, 30, 42 and 51 hpi. Cells were centrifuged, washed and pellets were processed for IFAs using established lab protocols [24].

### 4.3. Brefeldin A Treatment

*Plasmodium falciparum* (3D7) segmented schizont—infected erythrocytes were synchronized using 65% percoll, washed and returned to culture until merozoite reinvasion commenced. At 2 h post invasion (hpi), mature parasites were killed and ring-infected erythrocytes were synchronized using 5% sorbitol. In order to investigate trafficking of endogenous PfMC-2TM from the parasite to erythrocyte cytoplasm, synchronized parasites were divided into eight cultures. One culture was left as an untreated control, one culture was treated with the equivalent volume of ethanol used for solubilizing BFA and the remaining six cultures were treated with 10 µg/mL BFA at individual stages when parasites were at ring (2 hpi), trophozoite (30 hpi) and schizont (42 hpi) stages. Duplicate cultures were maintained at each stage. Ring stage parasites were treated with BFA and cultured for 16 h, trophozoites for 12 h and schizonts for 9 h. Following incubation with BFA, parasites were collected, centrifuged and pellets separated from supernatant. One set of cultures was collected after the drug treatment followed by 3 washes using RPMI-1640. Cells were mixed with complete media and cultures resumed for another 16 h for rings, 12 h for trophozoites and 6 h for schizonts to evaluate recovery of protein expression, translocation and viability of the culture. Infected erythrocytes were centrifuged, supernatants were discarded. Smears were prepared from the pellets for IFA. 

### 4.4. Permeabilization of Schizont-Infected Erythrocytes with Streptolysin O, Saponin and Trypsin

Infected erythrocytes were treated with saponin (0.02, 0.05, 0.09, 0.15 and 0.2%) and SLO (2, 4 and 8 units) for titration studies to determine optimum concentrations required for experiments. Saponin concentration at 0.09% in 1X PBS was added to infected erythrocytes and incubated for 15 min on ice in the presence of a cocktail of inhibitors [29,35]. SLO was activated using 10 mM DTT in 1X PBS before addition to infected erythrocytes. SLO treated cells were incubated at RT for 6 min [29], centrifuged to separate supernatant from pellets and the pellets washed in 1 X PBS. Infected erythrocytes were fixed with 0.5% formalin on ice for 30 min, followed by washing with 1X PBS. Late rings to early trophozoite-infected erythrocytes divided into four 100 µL aliquots were incubated with 0.09% saponin and 4 units SLO in 1X PBS. The first aliquot was SLO treatment alone. The second aliquot was treated with 1 mg/mL trypsin for 15 min at 37 °C after SLO treatment, the third aliquot was treated with 0.09% saponin on ice for 15 min following SLO treatment, and the fourth aliquot was treated with a combination of saponin and mg/mL trypsin after SLO treatment [29]. Smears were prepared for IFA as described [24].

### 4.5. PfMC-2TM Solubility Assays and Proteinase K Digestion of Infected Erythrocyte Membrane Ghosts

Membrane ghosts were prepared from predominantly mid schizont-infected erythrocytes (8% parasitemia) by hypotonic lysis in 10mM Tris pH 8 or 1:5 diluted RPMI in water [28], in the presence of a cocktail of protease inhibitor (Sigma-Aldrich, St. Louis, MO, USA). Briefly, 1 mL of infected and uninfected erythrocytes in separate tubes were mixed with 40 volumes of ice-cold lysis buffer, incubated for 15 min on ice and centrifuged for 30 min at 15,000 rpm at 4 °C. Membrane ghosts were collected and washed three times by centrifugation. Membrane ghosts were resuspended in 9 volumes of lysis buffer and stored at 4 °C. Four 36 µL aliquots of membrane ghosts were mixed either in 4 µL water, 4 µL 5% Triton X-100, 4 µL 50 mg/mL proteinase K, or 4 µL proteinase K and 5% Triton X-100 as described [26]. Tubes were incubated on ice for 1 h, with gentle mixing. Five µL of a protease inhibitor cocktail was added, followed by 6X electrophoresis sample buffer. Samples were boiled for 5 min for SDS-PAGE and Western blot analysis. Membrane ghosts were also divided into eight 100 µL aliquots and incubated with 100 µL each of 1% Triton X-100 in 10 mM Tris pH 8, 1mM EDTA (1), 8 M urea in 10 mM Tris pH 8, 1 mM EDTA (2), 100 µL 200 mM sodium carbonate, pH 11.5 (3), 100 µL 200 mM sodium carbonate, pH 11.5 mixed with ghosts after 4 cycles of freeze–thaw (4), 100 µL 1 mg/mL trypsin in 1X PBS (5), 100 µL mg/mL trypsin; 1 % Triton X-100 in 1 X PBS (6), 100 µL 10 mM Tris pH 8, 1 mM EDTA (7) and 100 µL of 1mM 10 mM Tris pH 8, 1mM EDTA (8). Tubes 1, 3, 4 and 7 were incubated for 30 min on ice; tubes 2 and 8 were incubated at room temperature for 30 min and tubes 5 and 6 were incubated at 37 °C for 30 min. Trypsin digestion was stopped by the addition of Soybean trypsin inhibitor at a final concentration of 2 mg/mL along with a cocktail of protease inhibitors. All tubes except sodium carbonate extractions were centrifuged at 13,000 rpm for 5 min. Supernatants were separated from pellets and added to fresh Eppendorf tubes. Sodium carbonate extractions were centrifuged in an Airfuge (Beckman Coulter, Brea, CA, USA) at 90,000 rpm for 40 min at room temperature. Supernatants were separated from pellets and added to fresh Eppendorf tubes. Pellets were washed once in ice-cold distilled water by centrifugation in a microfuge, for 5 min at 13,000 rpm. Supernatants and pellets from the eight tubes were solubilized in 6X electrophoresis sample buffer for SDS-PAGE and Western blot analysis. 

### 4.6. Antibodies

The following antibodies were used in immunofluorescence and confocal microscopy studies: pooled clones of monoclonal antibodies (Mabs) SP1C1 [55], peptide-specific rabbit antisera against three peptides from the PfMC-2TM protein encoded by gene PFA0680c (PF3D7_0114100) [24], Maurer’s clefts-specific antibodies; anti-REX1, and anti-MAHRP1 [13,56]. For the PVM, rabbit antibodies specific for SERA1 [55] and for *P. falciparum* RhopH3, antisera [57] were used. Monoclonal SP1C1 antibodies in spent culture supernatant were used undiluted. Rabbit antisera against the N and C termini of PfMC-2TM were diluted 1:25. Anti-ERD2 [34] (MR4, ATCC) was diluted at 1:1000, anti-SERA and anti-RhopH3 antibodies were used at 1:50 or 1:100. Anti-REX1 and anti-MAHRP1 were diluted 1:1000. Preimmune serum from mice and rabbits, were used as negative controls. Secondary antibodies conjugated to Alexa 488, Alexa 568, Alexa 594, Alexa 633 and Alexa 647 (Molecular Probes, Eugene, OR, USA), diluted 1:1000 in 1× PBS, were used to detect primary antibody reactivity.

### 4.7. SDS-PAGE and Western Blotting

Parasite extracts and the protein samples from the respective treatments were resolved in 6X SDS-PAGE buffer (Amresco, Solon, OH, USA). Following electrophoretic transfer of the separated proteins to nitrocellulose paper and blocking of unreacted sites with 2% milk, peptide-specific antibodies and control antibodies diluted in 1× TBS (Tris buffered saline) containing 2% nonfat dry milk (NFDM) were added to the nitrocellulose paper as primary antibodies. Each peptide antisera from different boosts for a single peptide were pooled and used for Western blotting. Following overnight incubation at 4 °C, the nitrocellulose papers were washed as described in [24] and incubated in anti-mouse (diluted 1:250) or anti-rabbit (diluted 1:5000) secondary antibodies conjugated to horseradish peroxidase and developed calorimetrically with 4-chloro-naphtol (Sigma-Aldrich, St. Louis, MO, USA).

### 4.8. Immunofluorescence and Confocal Microscopy

Immunofluorescence and confocal microscopy were performed as described in [24]. Microscopy was performed at the Cleveland Clinic Imaging Core, The Lerner Research Institute, Cleveland, OH, USA. Briefly, infected erythrocyte smears were fixed in cold methanol and acetone (1:1) for 5 min at -20 °C followed by incubation with rabbit and mouse polyclonal antibodies either individually or together in colocalization experiments. For colocalization assays, mouse and rabbit primary antibodies diluted as indicated above were mixed, and secondary antibodies directed to both species, conjugated to different colored fluorochromes, were mixed for detection of primary antibodies. Incubation with primary antibodies was carried out for 1 h; slides were washed three times with 1× PBS followed by incubation for 1 h with secondary mouse or rabbit antibodies conjugated to Alexa 488, Alexa 568, Alexa 594 and Alexa 633/647 (Molecular Probes). The smears were washed three times with 1 × PBS followed by one wash in distilled water supplemented with 8µg/mL bis-benzimide. Vectashield containing 4′, 6-diamidino-2-phenylindole (DAPI; Vector, Burlingame, CA, USA) or DAPI Fluoromount-G (Southern Biotech) was used to mount the slides. Images were collected using a Leica TCS-SP5II upright laser scanning confocal microscope (Leica Microsystems, GmbH, Wetzlar, Germany). In addition, an SP8 True Scanning Confocal (TCS) on a DMI8 inverted microscope was used to generate differential interference contrast (DIC) images.

## 5. Conclusions

Maurer’s clefts are parasite derived membranous structures produced in the erythrocyte cytoplasm as a result of cytoplasmic remodeling that occur immediately following merozoite invasion into the erythrocyte host cell. *P. falciparum* possesses multi-copy VSAs encoded by *var*, *RIFIN*, *STEVOR*, *SURFIN* and *PfMC-2TM* genes. VSAs participate in rosetting, knob formation and cytoadherance and are trafficked through the Maurer’s clefts alone or chaperoned by other parasite proteins that transiently associate with the MCs or are residents of the clefts. Permeabilization studies using SLO, saponin, digitonin and triton X-100 together with protease protection assays show that PfMC-2TM is present in the parasite cytoplasm, PV, MC and erythrocyte membrane. The high molecular weight rhoptry protein RhopH3 and the PV protein SERA1 were used as markers for compartment integrity to confirm the localization of PfMC-2TM and its distribution within the compartments. PfMC-2TM is expressed and exported using the classical secretory pathway into the erythrocyte cytoplasm for association with the MC in a topology that orients the N and C termini of the protein toward the erythrocyte cytoplasm as proposed in earlier studies [26]. PfMC-2TM is also trafficked to the erythrocyte membrane as an integral membrane protein with no change in solubility and resistant to triton and urea extraction. PfMC-2TM although integral in its association with the erythrocyte membrane was accessible to protease digestion. Taken together, our data shows the organization of the three TMs in the MC membrane with the sections of the protein associated with the MC membrane in a model that shows the topology of PfMC-2TM in the erythrocyte cytoplasm (Figure 9). The updated annotation of three TMs (PlasmodB) if accurate, may necessitate a change of the 2TM designation for the protein family with the topology model shown in Figure 9, if experimental data confirms the topology. Understanding the interactions of PfMC-2TM family members with RhopH3 and the erythrocyte membrane may yield novel insights regarding the immune and therapeutic potential of the PfMC-2TM family members.

## Figures and Tables

**Figure 1 pathogens-10-00431-f001:**
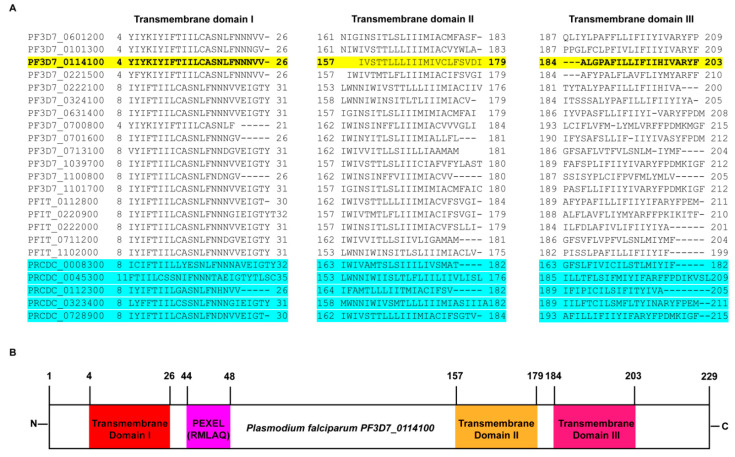
In silico characterization of PfMC-2TM. (**A**) PfMC-2TM multifamily protein sequence alignment, showing the three transmembrane domains. Antibodies specific to N- and C- termini and central portion of the protein PF3D7_0114100 (highlighted in yellow) were used in this study. The sequences highlighted in blue show the orthologs of PfMC-2TM in *P. rechinowi* the chimpanzee infecting species. (**B**) Schematic representation of the transmembrane domains of PfMC-2TM in *P. falciparum* and *P. reichenowi*.

**Figure 2 pathogens-10-00431-f002:**
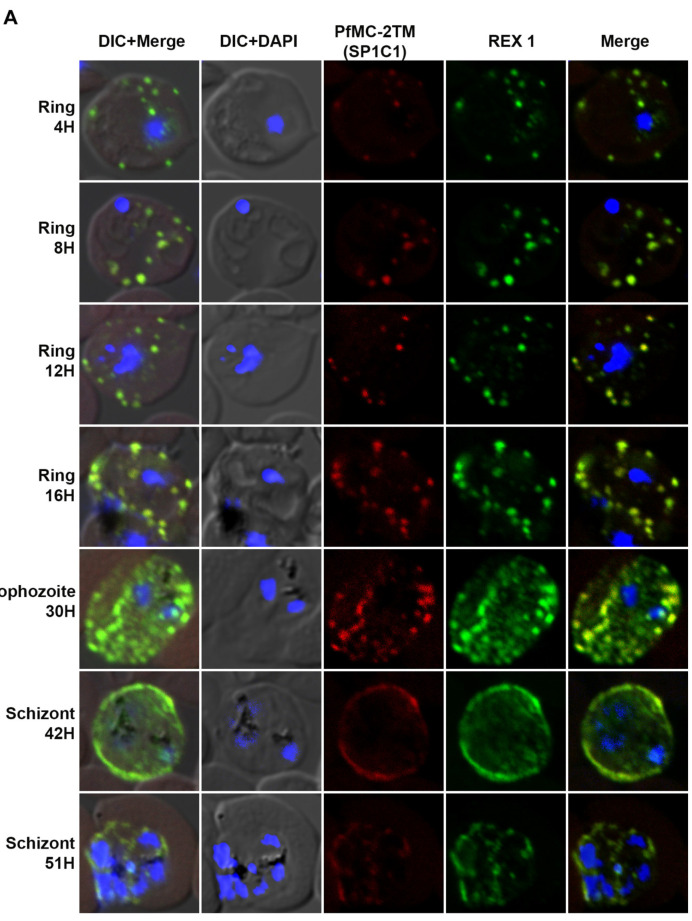
The localization of PfMC-2TM to Maurer’s clefts immediately after parasite invasion of red blood cell. (**A**) Stage specific expression and localization of PfMC-2TM in *P. falciparum* 3D7 infected erythrocytes was visualized using SP1C1 (mAb) and co-localized with REX1 (green) specific antibodies. Infected red blood cells were percoll synchronized and collected 4, 8, 12, 16, 30, 42 and 51 h post invasion of merozoites. Expression of PfMC-2TM (red) and co-localization with REX1 (green). (**B**) Stage specific expression and localization of RhopH3 and SERA 1 using Anti-SERA1 and anti-RhopH3 antisera were used for endogenous expression controls. (**C**) Negative control normal rabbit serum and normal mouse serum do not react with parasite proteins and uninfected red blood cells. Nuclei are stained with DAPI.

**Figure 3 pathogens-10-00431-f003:**
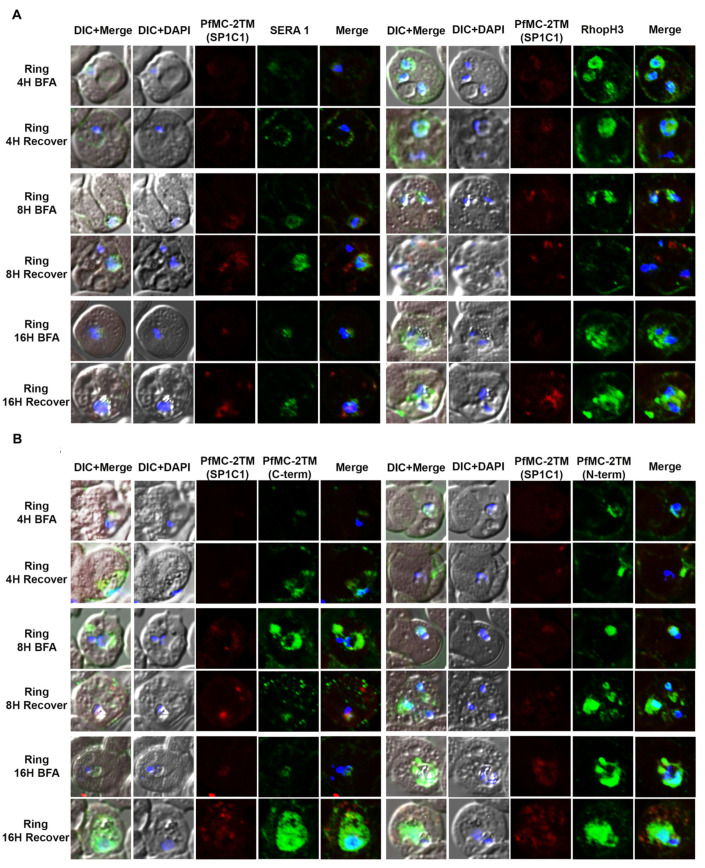
PfMC-2TM is exported via classical secretory pathway during the ring stage of *Plasmodium falciparum* (3D7) strain. Immediately after merozoite invasion, 3D7 ring stage parasites were treated with BFA for 16 h and timepoints were collected at 4, 8 and 16 h, respectively. The cells after respective timepoints were washed and recovered. (**A**). Monoclonal antibodies (mAb) SP1C1 specific for PfMC-2TM, anti-SERA1 specific for the parasitophorous vacuole protein (SERA1), and anti-RhopH3 (green) were used. (**B**) Antibodies SP1C1 specific for PfMC-2TM and antibodies specific for the C and N termini of PfMC-2TM were used. (**C**) Monoclonal antibody SP1C1 (red) specific for PfMC-2TM and antibodies specific for the MC resident proteins REX1 (green). MAHRP1 (green). (**D**) SP1C1 and antibody SP1A6 against the MC protein Pf130 were also incubated with anti-PfERD2. Briefly, PfMC-2TM (red), SERA1 (green), RhopH3 (green), REX1 (green) and PfERD2 (green) were visible inside the parasite in close proximity to the nucleus in Brefeldin-A treated cells. Removal of drug pressure shows translocation of PfMC-2TM into the cytoplasm and association with the MCs in the erythrocyte cytoplasm (Recover).

**Figure 4 pathogens-10-00431-f004:**
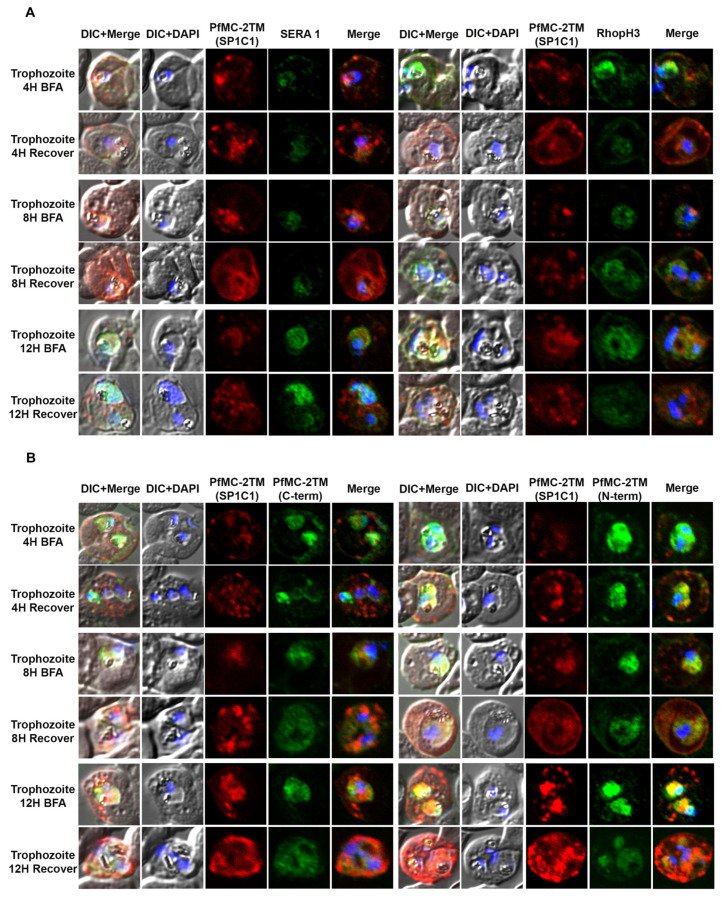
PfMC-2TM remains sensitive to Brefeldin-A during the trophozoite stage of *Plasmodium falciparum* 3D7 strain. Twenty hours old, early trophozoites were treated with BFA for 12 h and timepoints were collected at 4, 8 and 12 h, respectively. The cells after respective timepoints were washed and recovered. (**A**). Monoclonal antibody (mAb) SP1C1 specific for PfMC-2TM, anti-SERA1 specific for the parasitophorous vacuole protein (SERA1), and anti-RhopH3 (green) were used. (**B**) Antibodies SP1C1 specific for PfMC-2TM and antibodies specific for the C and N termini of PfMC-2TM were used. (**C**) Monoclonal antibody SP1C1 (red) specific for PfMC-2TM and antibodies specific for the MC resident proteins REX1 (green). MAHRP1 (green). (**D**) SP1C1 and antibody SP1A6 against the MC protein Pf130 were also incubated with anti-PfERD2. Briefly, PfMC-2TM (red), SERA1 (green), RhopH3 (green), REX1 (green) and PfERD2 (green) accumulated inside the parasite in close proximity to the nucleus in Brefeldin-A treated cells. Removal of drug pressure shows translocation of PfMC-2TM into the cytoplasm and association with the MCs in the erythrocyte cytoplasm (Recover).

**Figure 5 pathogens-10-00431-f005:**
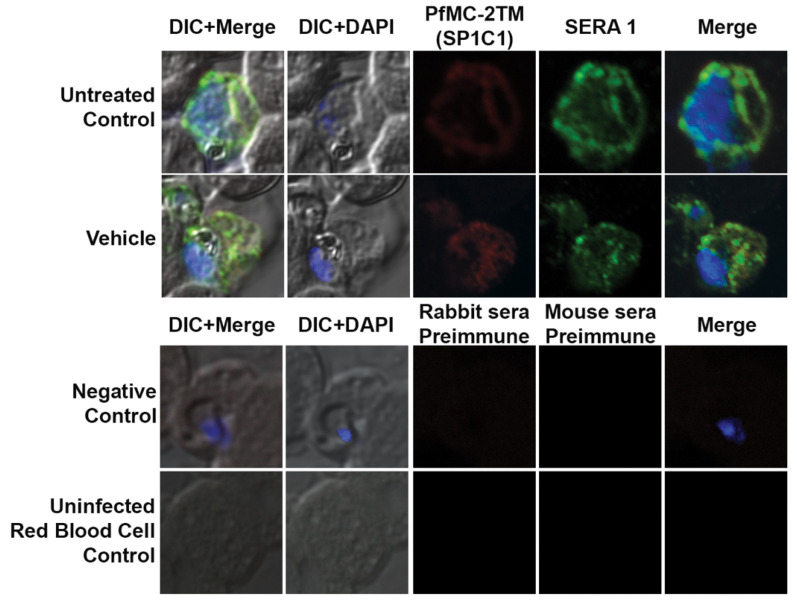
The untreated control and vehicle treated parasites were tested with α-PfMC-2TM (red) and α-REX1(Green) antibodies. The untreated and vehicle (ethanol) controls continued into the schizont stage of the parasite as multiple nuclei were visible inside the parasite and red blood cell. The parasites were viable in the vehicle treated cultures and the parasites progressed to the schizont stage. The normal rabbit serum and myeloma spent culture supernatant did not react with the parasite proteins. The antibodies α-PfMC-2TM (red) and α-REX1 (Green) did not react with the uninfected red blood cells. Negative control normal rabbit serum and normal mouse serum do not react with parasite proteins and uninfected red blood cells. Nuclei are stained with DAPI.

**Figure 6 pathogens-10-00431-f006:**
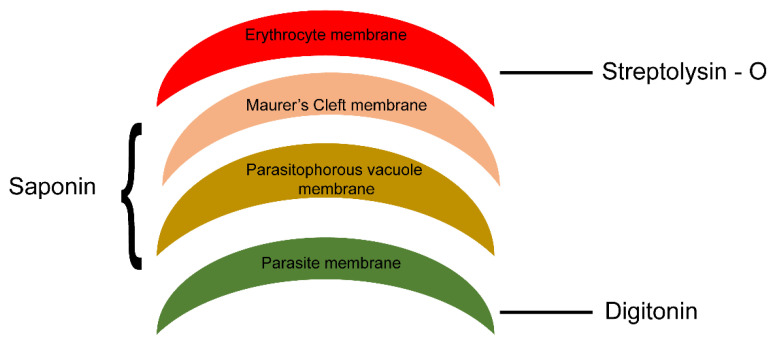
Schematic representation of membranes of different infected erythrocyte compartments permeabilized using SLO and detergent. Streptolysin (SLO) permeabilizes membrane of the RBC, saponin in combination with SLO permeabilize parasitophorous vacuole membrane and MC membrane and digitonin in combination with SLO permeabilizes parasite membrane leaving parasite organelles intact.

**Figure 7 pathogens-10-00431-f007:**
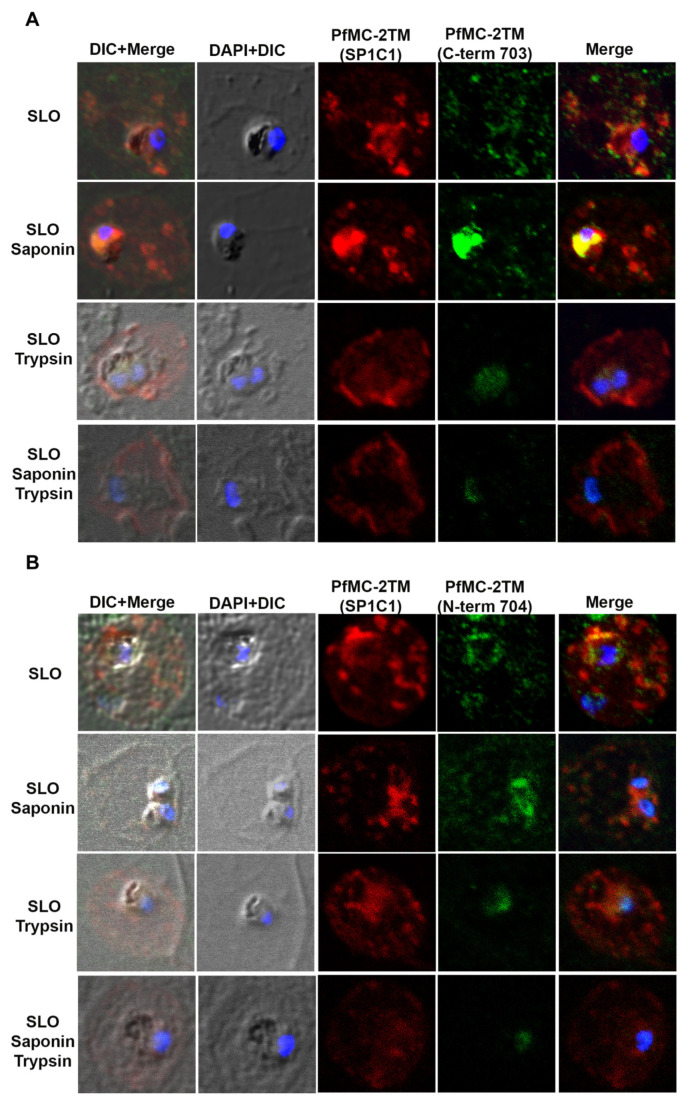
N and C termini of PfMC-2TM are exposed to erythrocyte cytoplasm. (**A**) SP1C1 antibodies were incubated with antibodies against the C-terminus of PfMC-2TM. (**B**) SP1C1 antibodies were incubated with antibodies against the N-terminus of PfMC 2TM. (**C**) SP1C1 antibodies were incubated with RhopH3 specific antibodies and SERA1 specific antibodies (7B). Reactivity of antibodies against RhopH3 and SERA1 were used to confirm the integrity of the compartments permeabilized by SLO and saponin. (**D**) Immunofluorescence of streptolysin O and saponin permeabilization of 3D7 infected erythrocytes followed by trypsin treatment. (**E**) SP1C1 antibodies were incubated along with antibodies against REX1. (**F**) Negative control normal rabbit serum and normal mouse serum do not react with parasite proteins and uninfected red blood cells. Nuclei are stained with DAPI.

**Figure 8 pathogens-10-00431-f008:**
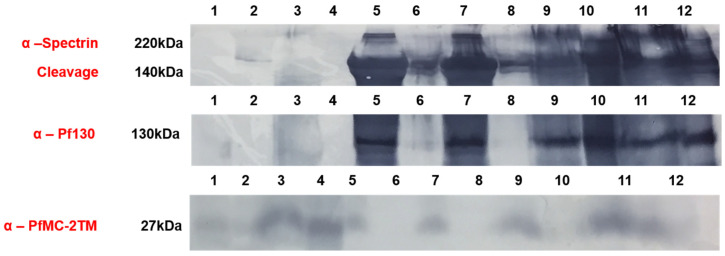
PfMC-2TM exhibits integral membrane protein properties and is accessible to proteases upon association with the iRBC membrane. Infected red blood cell membrane ghosts were subjected to alkaline sodium carbonate (Na_2_CO_3_) fractionation. Lane 1, infected RBC membrane ghost treated with Na_2_CO_3_ and mg/mL trypsin pellet; lane 2, infected RBC membrane ghost treated with Na_2_CO_3_ 1 mg/mL Trypsin (supernatant); lane 3, 3D7 parasite pellet treated with Na_2_CO_3_ and mg/mL trypsin pellet; lane 4, 3D7 parasite pellet treated with Na_2_CO_3_ and 1 mg/mL trypsin supernatant; lane 5, infected RBC membrane ghost treated with Na_2_CO_3_ pellet; lane 6, infected RBC membrane ghost treated with Na_2_CO_3_ supernatant; lane 7, infected RBC membrane ghost freeze—thaw and treated with Na_2_CO_3_ pellet; lane 8, infected RBC membrane ghost freeze—thaw and treated with Na_2_CO_3_ supernatant; lane 9, Parasites treated with Na_2_CO_3_, pellet; lane 10, Parasites treated with Na_2_CO_3_ supernatant; lane 11, Parasites (control) treated with Na_2_CO_3_ pellet and lane 12, Parasites (control) treated with Na_2_CO_3_, supernatant.

**Figure 9 pathogens-10-00431-f009:**
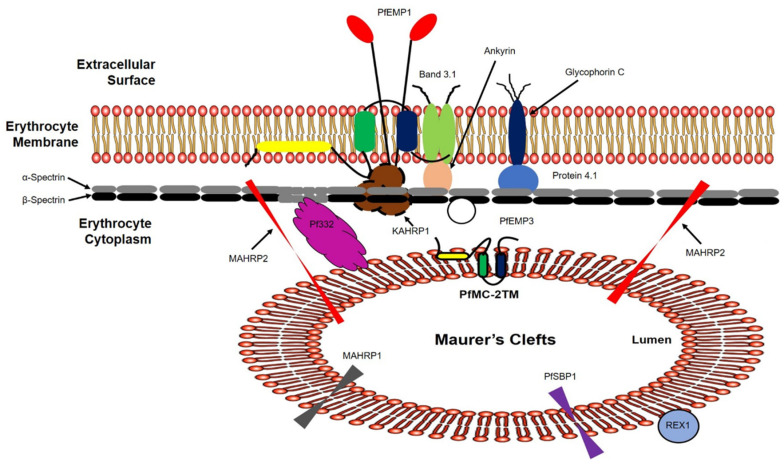
Schematic representation of PfMC-2TM mRNA and protein expression, and topological organization in the *P. falciparum* lifecycle. Based on the results in this study, we propose that PfMC-2TM is a resident Maurer’s cleft integral membrane protein. The central portion of the protein is protected in the Maurer’s cleft membrane with N and C termini exposed to the erythrocyte cytoplasm. Protease digestion studies further show that PfMC-2TM is associated with the erythrocyte membrane and is accessible to the proteinase K. We further propose that the protein associated as a monotopic integral membrane protein and interacts with KAHRP1 to assist in the anchoring PfEMP1 on to the erythrocyte surface. Alternately, iRBC surface treatment shows the loss of PfMC-2TM N-terminus signal using IFA. We further propose that the protein is transiently exposed on to the RBC surface. However, surface expression is inconclusive.

**Table 1 pathogens-10-00431-t001:** Time points followed for collecting the synchronized 3D7 in vitro cultures.

Time Point	Hours Post Infection (hpi)	Parasite Stage	Parasitemia Description
**1**	0–4	Segmenters Rings	Shizonts (1%) Rings (4%)
**2**	5–8	Rings	Rings (5 %) Schizonts (0.1%)
**3**	9–12	Rings	Rings (6 %) Schizonts (<0.1%)
**4**	13–16	Rings	5% Rings 1% Early trophozoites
**5**	18–30	Late rings Early trophozoites	0.5% Late Rings 5–6% Trophozoites
**6**	31–40	Late trophozoites Early schizonts	5% Early schizonts 1.5% Late trophozoites
**7**	42–51	Schizonts Segmenters	5 % Schizonts
**8**	51–3 (second cycle)	Segmenters Rings	3% rings 5% Segmented schizont

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
