# Peer review of "Trafficking and Association of Plasmodium falciparum MC-2TM with the Maurer’s Clefts"

_pathogens, 2021, doi:10.3390/pathogens10040431_

Round 1

Reviewer 1 Report

The manuscript entitled “Trafficking and associating of Plasmodium falciparum MC-2TM with the Maurer’s clefts” is interesting. I suggest Authors to give more attention to the following remarks point-by-point:

  1. 1 is unreadable
  2. 6A should be a separate figure
  3. The title of all figure are too long

Author Response

Hello,

We thank the reviewer for their time in carefully reading our manuscript. The suggestions and comments are addressed in the manuscript.

The manuscript entitled “Trafficking and associating of Plasmodium falciparum MC-2TM with the Maurer’s clefts” is interesting. I suggest Authors give more attention to the following remarks point-by-point:

1. Figure 1 is unreadable

The figure has been relabeled for clarity.

2. Figure 6A should be a separate figure

Figure 6A is now a separate figure labeled as Figure 6

3. The title of all figures are too long

Titles and figure legends have been shortened for all figures.

Reviewer 2 Report

This manuscript is an in-depth investigation of the Plasmodium falciparum multi-gene family protein, PfMC-2TM. The first goal is to investigate the stage-specific expression and translocation into the erythrocyte membrane. The second goal is to attempt to clarify the topologic organization of the protein in both the Maurer’s cleft and the erythrocyte membrane. Both of these goals are needed in the field as there has been substantial controversy and confusion regarding these issues. The first goal is accomplished nicely with extensive immunofluorescence studies using well defined antibodies and confocal microscopy. The second goal is less clearly accomplished. Extensive differences between the findings in this study and previous work (especially Bachmann et al, 2015) confuse the issue. These differences start with the fact that the protein appears to have been re-annotated to contain three rather than two TM domains, and continue to differences in experimental results between the two studies that are difficult to explain. There also appears to be results in this report (ie trypsin susceptibility of membrane ghosts) that are difficult to explain with the author’s current hypothesis. However, the work is carefully performed and (usually) well-explained. The authors are forthcoming that all of the answers are not yet ‘in’. The data need to be published and available to the scientific community so that further work can be performed in this important area. My specific comments are as follows: Major Starting with Section 2.6 the manuscript becomes less clear. Prior to this the authors do a very nice job of explaining the scientific premise for each of their experiments – even including a figure to help the reader understand the different effects of various permeabilizing agents. Equal care should be taken from this section forward, including an explanation of the different solubilizing agents, and the characteristics of cell ghosts and the different lysis procedures. In addition, phrases such as “PfMC-2TM is facing the cytoplasmic side of the erythrocyte membrane” are confusing. Which portion of the protein is facing the cytoplasmic side, N-terminal? C-terminal? central core? Minor In the final paragraph of the Introduction - the sentence starting with “Using” needs a verb. Figure 2 Legend. Part A seems as if there is some redundancy. Same for section (B) why are hours post invasion listed twice (4,8,12 and 16 first and then 30,42, and 51 later)? Figure 3A second to the last row should be “12H BFA” rather than “16H BFA”. Section 2.4: It is unclear what age (hpi) parasites were used in these experiments. Entirely speculative: Do the authors suggest changing the name to PfMC-3TM?

Author Response

My specific comments are as follows:

Major Starting with Section 2.6 the manuscript becomes less clear. Prior to this the authors do a very nice job of explaining the scientific premise for each of their experiments – even including a figure to help the reader understand the different effects of various permeabilizing agents.

In order to clarify the experiments performed to address our second goal. We have added a short description of the effects of each solubilization agent used, including the rationale for using each one in the section 2.6 of the manuscript as follows:We investigated the solubility properties of PfMC2TM using Triton X-100, alkaline sodium carbonate, urea and EDTA. Triton X-100 solubilizes peripheral and transmembrane proteins, sodium carbonate at pH 11 extracts peripheral membrane proteins and leaves transmembrane proteins intact in the pellet, 8M urea is a chaotropic reagent that solubilizes and disrupts protein complexes and EDTA chelates divalent cations destabilizing membrane proteins. The different reagents were used to treat infected red membrane ghosts obtained after hypotonic lysis of red blood cells as well as parasite pellets in the presence of absence of proteases. PfMC2TM would be accessible to protease treatment in domains exposed to the erythrocyte cytoplasm. Retention of PfMC2TM in the infected membrane ghosts following alkaline carbonate extraction and treatment with urea would indicate that the protein retained its integral membrane properties and was insoluble, respectively”.

Equal care should be taken from this section forward, including an explanation of the different solubilizing agents, and the characteristics of cell ghosts and the different lysis procedures.

Please see the response above. We have clarified our use of solubilizing agents for determining the topology of PfMC-2TM.

In addition, phrases such as “PfMC-2TM is facing the cytoplasmic side of the erythrocyte membrane” are confusing.

We changed the phrase, “PfMC-2TM is facing the cytoplasmic side of the erythrocyte membrane” with “PfMC-2TM is exposed to the erythrocyte cytoplasm”.

Which portion of the protein is facing the cytoplasmic side, N-terminal? C-terminal? central core? Minor In the final paragraph of the Introduction - the sentence starting with “Using” needs a verb. Figure 2 Legend. Part A seems as if there is some redundancy.

The N- and C-termini of PfMC-2TM are exposed to the erythrocyte cytoplasm. The transmembrane regions of the protein are in the MC and RBC membrane.

Same for section (B) why are hours post invasion listed twice (4,8,12 and 16 first and then 30,42, and 51 later)? Figure 3A second to the last row should be “12H BFA” rather than “16H BFA”.

This section has been revised to show that Parasite-infected erythrocytes were collected at 4, 8, 12, 16, 30, 42 and 51 hpi.”

Section 2.4: It is unclear what age (hpi) parasites were used in these experiments.

The parasites used were 28-30 hpi.

Entirely speculative: Do the authors suggest changing the name to PfMC-3TM?

Additional studies are needed to investigate the three transmembrane domains in order to understand the role of the domains to the function of PfMC-2TM. A change of the 2TM designation for the protein to PfMC-3TM will be determined by additional data.